# Effects of Changes to Architectural Elements on Human Relaxation-Arousal Responses: Based on VR and EEG

**DOI:** 10.3390/ijerph18084305

**Published:** 2021-04-19

**Authors:** Sanghee Kim, Hyejin Park, Seungyeon Choo

**Affiliations:** School of Architecture, Kyungpook National University, 80, Daehak-ro, Buk-gu, Daegu 41566, Korea; sangheekim@knu.ac.kr (S.K.); phj8598@knu.ac.kr (H.P.)

**Keywords:** architectural elements, EEG, VR, relaxation-arousal reaction, healing space

## Abstract

This study combines electroencephalogram (EEG) with virtual reality (VR) technologies to measure the EEG responses of users experiencing changes to architectural elements. We analyze the ratio of alpha to beta waves (RAB) indicators to determine the pre- and poststimulation changes. In our methodology, thirty-three females experience using private rooms in a postpartum care center participated in the experiment. Their brain waves are measured while they are experiencing the VR space of a private room in a postpartum care center. Three architectural elements (i.e., aspect ratio of space, ceiling height, and window ratio) are varied in the VR space. In addition, a self-report questionnaire is administered to examine whether the responses are consistent with the results of the EEG response analysis. As a result, statistically significant differences (*p* < 0.05) are observed in the changes in the RAB indicator values of the pre- and poststimulation EEG while the subjects are experiencing the VR space where the architectural elements are varied. That is, the effects of the changes to architectural elements on users’ relaxation-arousal responses are statistically verified. Notably, in all the RAB indicator values where significant differences are observed, the poststimulation RAB decreases in comparison to the prestimulus ratios, which is indicative of the arousal response. However, the arousal levels vary across the architectural elements, which implies it would be possible to find out the elements that could induce less arousal response using the proposed method. Moreover, following the experience in the VR space, certain lobes of the brain (F4 and P3 EEG channels) show statistically significant differences in the relaxation-arousal responses. Unlike previous studies, which measured users’ physiological responses to abstract and primordial spatial elements, this study extends the boundaries of the literature by applying the architectural elements applicable to design in practice.

## 1. Introduction

### 1.1. Background and Objectives

Architecture and built environments are conducive to healing, should the relevant spatial environment be duly considered. Although the environment itself cannot treat diseases, the natural environment presented in architecture and built environments may add to positive emotions [1].

Moreover, quite a few studies demonstrated that the physical environment itself is related to users’ physical, and mental health, cognitive performance, and behavior, and that mental health can be assessed in terms of the awareness of the environment through all human senses [2]. In other words, the physical environment may act as the important media underpinning people’s emotional and psychological well-being [3].

As one of the most influential scholars in the architectural design of healthcare buildings, Roger S. Ulrich measured a method of substantially reducing the postoperative recovery time in hospitals for the first time [4]. Although modernist architects worked on the theories about the relationship between architectural design and health, it was Roger S. Ulrich who quantified the effects of patients’ environment on their healing process for the first time [5]. It was not until the late 20th century when the relevance between the brain and the immune system was found critical for health maintenance and when the potential effects of architectural space on health were scientifically explored, as exemplified by Roger S. Ulrich (1984) [5]. The foregoing research efforts have expanded the horizon of the studies on the methods of designing the physical space inducing comfort or relaxation beyond psychological or observational surveys.

In the same vein, the decades-long paradigm in the field of environmental psychology needs to be briefly reviewed. In the 1920s, as an architect, Mayer felt the need for interactions between architects and psychologists, and took the initiative in asserting that architecture should be planned considering human physiology and psychology, which triggered the interdisciplinary collaborations between architecture and environmental psychology. In the 1950s, architects came to collaborate with behavioral scientists. In 1961, a conference held in Utah in the U.S. was focused on the relationship between the architectural design of psychiatric wards and the evidence of treatment and recovery and began to use the term ‘environmental psychology’. In the 1970s, multidisciplinary connections between environmental design, psychology, and sociology started in earnest. In the 1980s, the scope of research was broadened, ranging from environmental perception, cognition, and attitude to the meaning of the environment, while more systematic research was conducted to apply the human psychology of preference and behavior to environmental design

In 2004, architecture collaborated with neuroscience. The launch of the Academy of Neuroscience for Architecture by neuroscientists and architects in San Diego in the U.S. invigorated the research on humans living in artificial architectural structures, engendering a new field of ‘neuroarchitecture.’ Largely, neuroarchitecture is based on two assumptions. One is that the ‘human cognitive thinking process’ is directly and indirectly influenced by spatial elements, while the other is that the ‘cognitive influence’ of space on humans is physically observable and measurable [6]. These assumptions became of central interest to this study.

In brief, research on the psychological, physical, and behavioral effects of architectural space on humans has drawn attention, whilst the design based on systematic and objective study results and grounds, not intuitive and subjective decisions, is regarded as the requisite for creating a beneficial environment [7]. The assumptions and study results underlying environmental psychology and neuroarchitecture have altered the architectural design practice, which used to rely on experience and intuition [6].

In the process, neuroscientists played some roles by providing new tools and methods for investigating the accurate correlation between environmental stimuli and human emotions. Among the various methods, the electroencephalogram (EEG) has been used by researchers in the design and architectural environments since the 1980s to gain new insight into how people perceive the characteristics of built environments and design [8]. Moreover, a number of authors have performed experiments on EEG measurements in combination with virtual reality (VR) technology with a view to addressing the limitations of the EEG-related literature and pushing the boundaries of research [8,9,10,11].

Given the physiological effects of architectural space on humans are measurable and grounded on scientific evidence as suggested in the literature, it would be possible to establish the elements of spatial awareness that exert positive effects on humans.

Thus, in this study, we established the effects of varied architectural elements on users’ relaxation-arousal responses by measuring the EEG response to the cognitive effects exerted by the space represented in immersive VR. We used the ratio of alpha to beta waves (RAB) indicators to analyze relaxation-arousal responses and performed the Wilcox signed-rank test to analyze any statistically significant differences in the pre/poststimulation EEG data. Based on the results, we proposed some scientific reference data applicable to the healing space design that could induce comfort for humans, as well as the physiological methodology combining the EEG with the latest VR technology.

### 1.2. Scope

In this study, we modeled and represented the private rooms in a postpartum care center in immersive VR. In Korea, mothers usually opt for the private rooms in postpartum care centers for 2–4 weeks shortly after giving birth. Over 85% of mothers experience postpartum blues and mood disorders and suffer anxiety, restlessness, and emotional fluctuations [12], the private rooms in postpartum care centers need to be spatially designed in a way that induces mothers’ psychological stability.

The participants in this study were limited to females who had the experience of using private rooms in postpartum care centers. The primary purpose of this study was, through analyzing EEG response, to present a scientific rationales to be considered in the actual space design. It is reasonable, therefore, to limit the participants to females who had experiences of using that space rather than enrolling the general population to ensure that the results are relevant for designing a private room.

Therefore, we applied different types of aspect ratios of space, ceiling heights, and window ratios to the private rooms in the postpartum care center as the architectural elements relevant to the perceived sense of space.

This study was focused on verifying any statistically significant differences in the pre/poststimulation RAB indicators in response to the different types of architectural elements in equal-size VR spaces, and examining if certain lobes of the brain were involved in the relaxation-arousal responses. We set up the following null hypotheses for statistical verification.

**Hypothesis** **1.***The experience in a VR space varying with different types of architectural elements will lead to no significant differences in the EEG-based relaxation-arousal responses*.

**Hypothesis** **2.***No particular lobes of the brain will be involved in the EEG-based relaxation-arousal responses to the experience in a VR space varying with different types of architectural elements*.

## 2. Application of VR-EEG in Architectural Research

### 2.1. Combining the Need for Virtual Reality with VR-EEG Technology

The client who requested the space design or actual user wants to experience the space before completing construction. Although 2D or 3D images are generally used for such clients to experience the space indirectly, it is often difficult for the clients to fully grasp the sensation about space according to movement. The VR environment, to overcome these limitations, allows us to experience various sensations according to movements [13]. 

The VR also has the advantage of building an environment pursued by researchers and controlling variables. It is consequently possible to measure individuals’ emotional response in an environment controlled by researchers [14].

In addition, one of the most prominent advantages of VR is that it maximizes the immersion more easily compared to the real environment and that the standardization is possible with relatively low cost. These make the VR essential in the field of construction [15,16]. 

Using the advantages of VR mentioned earlier, we use Miguel et al. [11]. Researchers propose a VR space that reproduces the actual dyeing treatment room as an alternative to the dyeing treatment room for stress relief. In this work, VR-related costs are much lower than those associated with standard hue investment and maintenance, and can be implemented in commercial devices, increasing the availability and convenience of pigmentation.

According to various literature using VR, virtual reality has been successfully applied in many fields, including phobia$ [17,18] and other disorders [19,20], rehabilitation [21,22,23], and areas, such as learning [24,25,26], industry [27], and marketing [28]. In most of the cited studies, the use of VR showed significant improvements, compared to traditional approaches [11]. Other studies have shown that VR can help deal with stress [29,30,31].

These studies show that VR can be a very useful technique for developing techniques for designing healing spaces, and VR has already been recognized for its achievements in various research fields.

Meanwhile, the immersion technology is developing at a significant speed also in its commercial use, and it is expected, together with augmented reality, to be used increasingly in the product and service fields in the future [32].

Recently, VR technology is trying to expand the scope of research by combining with EEG technology. VR has been combined with mobile brain/body imaging (MoBI), which synchronously records movement and brain dynamics during active exploration of the virtual environment [33]. Furthermore, prior work has utilized EEG techniques when performing evaluations of virtual reality environments [8,9,11].

This suggests that the EEG, combined with VR technology, is a scientific and objective method to replace existing inaccurate and self-report evaluation methods [11]. The aforementioned research results and social background will answer why VR technology should be used in architectural space design.

### 2.2. A Study on the Utilization of EEG in Architecture and Measurement of Relaxation-Awakening Response

The recent advancements in neuroscientific methods allowed us to study human cognition and affective states in the construction field [14]. Among these academic disciplines, cognitive neuroscience is a subfield that is integrated into the construction field to study design. The cognitive neuroscience approach may be a useful method to explore human response to the design of the built environment. Various methods have been tested and implemented for past decades, and functional Magnetic Resonance Imaging (fMRI) and electroencephalography (EEG) have been used as practical tools in applying cognitive neuroscience [8].

Among them, EEG, a noninvasive method to investigate brain activity, has advantages of being safe, easy operation, and low cost. It also allows real-time observation of brain activity and personal use. The data obtained from EEG experiment is likely to be used commercially and personally in the open market [34].

For the methods to perceive the built environment and design elements, the signal detected from the EEG provides new insights into the results of this study. Researchers in the fields of design and built environment have used extensively the EEG since the 1980s [8]. However, there are not many studies that objectively clarify the impact of architectural space on people’s emotions and behaviors, while it is important to find mental and physical effective design methods in the field of architectural design [35]. Recently, the trend has been to overcome these limitations and quantitatively measure the impact on human emotions and behavior in architectural space, and existing prior studies have been conducted on the following topics.

Studies of changes in brain waves in environmental and spatial stimuli have been conducted on a variety of topics, including studies of human emotions and relationships with architectural space [36,37,38], stress and anxiety [11,39,40], studies of positive and negative emotions [41,42], concentration and attention [43,44], and cognitive and emotional processing [9,14,45]. These studies have demonstrated that EEG is an appropriate tool for achieving research objectives.

The four frequency bands of EEG signals are important to analyze data and represent different brain activities. First, the Delta band (0.5–4 Hz) is activated in the state of deep sleep and waking, the Theta band (4–8 Hz) in the state of excitement, deep meditation, creative inspiration, and flow of awareness, the Alpha band (8–12 Hz) in the state of relaxed awareness free from being attentive, and the Beta band (12–30 Hz) in the state of active thinking and attention and problem-solving [26]. The relaxation-awakening reaction of brain waves is mainly related to alpha and beta waves. The alpha wave is classified into fast and slow alpha waves. The former is associated with an idling state, and the latter is associated with a state of relaxation, eye closing, calm, and resting. The reduced alpha save suggests a shift in brain function to a more alert state of mind of beta wave [46].

The beta wave is observed mainly in the state of active thinking and concentration, where the mid beta wave, associated with high alert and low beta waves, is observed in the state of resting and thinking [47,48]. Among them are alpha and beta waves, and the ratio of alpha waves to beta waves (RAB: alpha/beta ratios) shows a close relationship with behavior. Participation in cognitive tasks significantly reduces the proportion of alpha/beta power [49]. Studies that utilize the characteristics of these brain wave bands and the alpha and beta wave frequencies to measure the relaxation-awakening response of brain waves to stimuli include: First, Tee et al. [50] showed that the proportion of alpha and beta waves is negatively correlated as stress increases. That is, the ratio of alpha to beta waves indicates that the data characteristics of brain waves can be distinguished for stress evaluation. Emanuel et al. [51] measure changes in the signal amplitude (Alpha and Beta, alpha/beta) of brain waves under uncomfortable environmental conditions with respect to thermal environments, resulting in cognitive performance and thermal environment. Sosiedka et al. [48] calculated by dividing the slow wave frequency by the fast wave frequency for the emotional activation effect on olfactory stimuli. In other words, we analyze the changes in theta/ratio and alpha/beta ratios to prove that they are characteristic of subjective emotion evaluation and brain activation. Benjamin et al. [49] established that the alpha/beta ratio decreases as the stimulus of certain information increases in visual perception, auditory perception, and visual memory recovery tasks. Hwang [52] analyzed the awakening effects of brain waves on color and shape stimuli with the ratio of alpha waves to beta waves. Kim et al. [53] then provided an objective basis for applying the relaxation-awakening state of brain waves to real-world lighting production by measuring the ratio (RAB) of alpha waves to beta waves.

This work extends the subject in detail to propose a method for obtaining more practical objective data in the field of architectural design.

In other words, we disaggregate the architectural elements that stimulate brainwave reactions and differentiate them from prior work in that we analyze them through RAB indicators that are involved in relaxation-awareness rather than interpreting emerging brainwaves for stimuli as they are.

### 2.3. Cognitive and Physiological Responses to Architectural Elements

The previous studies on the users’ cognitive and physiological responses to the architectural elements were reviewed to develop the hypotheses of this study. Arthur E. S. (2011) adopted a VR street model and found that the range of area perceived may be expanded b modifying the space shape and aspect ratio even if the actual space size is fixed. In addition, a study on indoor hotel room space proved that the ratio of width and length (1:1, 1:2, and 1:9) had a stronger effect on the space area perceived by users compared to horizontal area and height factor [54].

Subklew, F. (2009) [55], using a 3D modeling for virtual commercial space, examined the effect of ceiling height on users’ recognition of control and their assessment for store and Joan M. and Rui, Z. (2007) [56], similar subjects, proved that the change in ceiling height may have effects on consumers’ information processing and behavior. Vartanian et al. (2015) [57], using functional magnetic resonance imaging (fMRI), examined the effect of ceiling height on the aesthetic assessment for architectural design and avoidance determination and reported that the rooms with higher ceilings are likely to be assessed as a beautiful one. Nikravan M. (2014) [58], through a questionnaire survey, examined the effect of ceiling height on the comfortableness of the people in the experimental room with adjustable ceiling height.

Ulrich, RS (1984) tested whether the natural scenery seen out of the window calms the inpatients’ mind and has a positive effect on their health recovery through the reduction of hospitalization stress and reported its positive effect on healing [59]. Barbara M. (2006) verified that the shape of the window had effects on the perception of room size in the room with a small area [60].

## 3. Experiment and Analysis

### 3.1. Hypotheses

This study was focused on verifying any statistically significant differences in the pre/poststimulation RAB indicators in response to the different types of architectural elements in equal-size VR spaces, and examining if certain lobes of the brain were involved in the relaxation-arousal responses. The following null hypotheses for statistical verification were developed based on the previous studies on cognitive and physiological responses to architectural elements (see Section 1.2).

### 3.2. Participants

The participants in this experiment utilizing the VR-EEG were females who had used private rooms in postpartum care centers. After obtaining the Kyungpook University Institutional Review Board (IRB) approval (KNU-2020-0096) on the study plan, we publicly recruited the voluntary female participants by posting the study methods and objectives on the internet. It took six days to recruit the sample. Larson and Carbine con-ducted a systematic review on the sample sizes used in human electrophysiology (EEG and ERP), and stated that the sample size calculation had been hardly reported in current clinical human electrophysiology literature and that the sample size of particular research was relatively small, in the range of 7–26 subjects per group [55]. That is, as any particular sample size is not specified for EEG experiments, we recruited 35 participants as the minimum sample size for the statistical analysis. Excluding two persons who were found to be on medication in the pretest interview, 33 participants were included in the experiment.

The participants met the inclusion criteria that required no history of brain and mental diseases, normal blood pressure, no heart diseases, no anxiety in closed space, no medication for treatment, no insomnia or sleep-related diseases, and no eyeglasses because they had to wear the VR equipment. Moreover, they were instructed to refrain from drugs and caffeine from the day before the start of the experiment, and to take a good rest.

Prior to the experiment, the participants filled in the questionnaire, which included the general information and self-assessment of stress. The self-diagnose of stress was intended to determine their stress levels on the day of the experiment, since extreme stress or any wide gaps in stress levels among the participants could affect their EEG response. Moreover, in that their EEG response could not be deemed as attributable to stimuli if they were under stress, we examined the stress levels of the participants to increase the accuracy of the experimental results. For the self-diagnosis stress test, we used the PWI-SF (PWI-Short Form) among the PWI (Psychological Wellbeing Index), which was adapted for Koreans by Jang (1993) from the GHQ (General Health Questionnaires)-60 [61].

As for the stress scores, 27 and 6 participants out of 33 were classified into the potential stress group and the healthy group, respectively, which indicated relatively modest stress levels. Moreover, no substantial gaps in stress levels were found among the participants. An independent sample t-test was conducted to determine the homogeneity of the two groups before the stimulation between the potential stress group and the health group occurred. As a result, it was found that the background brain waves of all electrode channels were greater than the significant level (*p* < 0.05), and there was no difference between groups in Table 1. The general characteristics of the participants are shown in Table 2.

### 3.3. Selection of Architectural Elements and Production of Visual Stimuli

#### 3.3.1. Grounds for Architectural Elements Selected

We set up three architectural elements which were deemed to relate to the perceived sense of space and influence the EEG-based relaxation-arousal responses on the grounds specified below.

First, the aspect ratio of space means the ratio between the length and the width of a plane. Since architectural or spatial structures carry arithmetic and geometric attributes, proportionality serves as a stimulating factor that leads to the scales of dimensions and shapes or the spatial perception and cognition. Therefore, how people feel about an architectural space may vary with the proportionality between the architectural elements [62]. Indeed, architects will secure highly useful source data when they design buildings, should they first establish whether there are differences in users’ EEG-based responses to longer and wider planes when the areas of both are equal and how they differ. We set up two (Types A and B) aspect ratios of space by referring to the status of real-world postpartum care centers. The aspect ratio of Type A was 1:1.6, where the space was long seen from the door, while the aspect ratio of Type B was 1.6:1, where the space was wide.

Second, the ceiling height is another architectural element referring to the height from the floor finish to the ceiling finish. The ceiling height is one of the important elements used to measure the superiority of the indoor environment, in that differences in ceiling heights may serve as the indicators of whether people feel cramped or capacious in an architectural space [63]. The Building Act (≥2.1 m) and the Housing Act (≥2.2 m) of Korea stipulate only the lower limits of ceiling heights. We determined the reference point of the ceiling height applied in this study by reviewing the literature and the design trends of the country’s apartment houses. Bae D. [63] proposed ≥2.375 m and ≥2.325 m as an ideal and minimum ceiling heights of living rooms in apartment houses, respectively, based on survey results. Nikravan M. [58] carried out a survey on ceiling heights of 2.1 m–4 m and proved 2.7–2.8 m to be comfortable ceiling heights in general residential cities. Joan et al. [56] had their participants solve some problems that required creativity and concentration in 3 buildings whose ceiling heights were 2.4 m, 2.7 m, and 3 m each, and demonstrated higher ceilings activated creative thinking, whereas lower ceilings facilitated concentration. Currently, in Korea, a specialized design for apartment houses with living room ceiling heights raised to 3 m from the existing 2.3 m–2.4 m to maximize openness and enhance spatial efficiency is drawing attention [64]. Based on the literature and real-world designs, we set up three types of ceiling heights, viz. 2.3 m, 2.7 m, and 3 m.

Third, the window ratio is the other architectural element, which was determined using the following formula: (window area/(exterior envelope wall area + window area)) × 100%). As mentioned in the introduction section, windows are architectural elements whose healing effects have been proved. Moreover, the WELL Building Standard (US, 2014), a certification system for rating architectural environments focused on human health, describes windows as a WELL strategic content in relation to air, light environment, agreeableness, and mental health items, all of which come down to ventilation, windows, natural daylighting, natural ventilation environment and connectedness with nature [65].

In this study, we examined the EEG-based relaxation-arousal responses to the window ratio, which is essential element involving the visual openness, visibility, view, and natural daylighting, among other design details of windows for a healing environment. We set up five types of window ratios by referring to Kim et al. [66], i.e., 20%, 40%, 60%, 80%, and 100%.

#### 3.3.2. Production of Visual Stimuli in VR Space

We applied a fixed internal area of 13 to the private room represented through the VR device by referring to the room sizes in M Postpartum Care Center in Busan. The area does not include the bathroom installed in each room. Moreover, the bathroom layout, entrance hall, furnishings, and view from the window of the postpartum care center were represented as is. We placed the window on the wall facing the entrance door. We created the VR space using Revit Architecture, and drew the floor plan, elevation, and ceiling plan before making the basic 3D model. We imported the basic 3D model into Twinmotion to produce the 360° 3D images to be implemented on the VR device. Twinmotion is a real-time immersive 3D architecture visualization solution for producing high-quality images [67]. We set up the daylight coming through the window at 13 o’clock. To represent the view from the window, we mapped the photos of the landscape seen from the postpartum care center. In total, we produced 30 360° 3D images for the VR space of the room in the postpartum care center (2 aspect ratios of space × 3 ceiling heights × 5 window ratios = 30). The two types of floor plan (Figure 1) and the featured image (Figure 2) are shown below.

### 3.4. Tools Used in the Experiment and EEG Data Analysis Indicators

We used SMI’s Eye Tracking HMD based on HTC Vive as the VR device, and the wired and wireless dry 24-channel EEG DSI-24 (Wearable Sensing, City, State Abbreviation, United States of America) in the experiment (Figure 3). As the EEG data analysis indicators, we used the alpha and beta wave frequencies, which are known to be involved in stress and relaxation-arousal responses. Particularly, we analyzed the RAB (alpha/beta ratios) as the indicators of the EEG-based response to the visual stimuli in the VR space. The RAB indicator values presented in this paper are the differences calculated by subtracting the prestimulation RAB indicators extracted from the background EEG from the RAB indicator values to the stimuli in the VR space (Figure 3).

### 3.5. Experimental Environment and Process

#### 3.5.1. Experimental Environment

We conducted the experiment in a lecture room at Kyungpook University for four days, from 29 October to 1 November 2020 (10:00–16:00 daily). We partially used the lecture room, which was spacious and best for noise control, for fear that the participants wearing the VR-EEG equipment might feel cramped in our relatively smaller lab (Figure 4). We checked if the temperature and humidity were ideal before starting the experiment. Particularly, we made sure the participants saw the view through the large window facing them while they were seated to help them feel stable physically and psychologically before and after the EEG measurement.

#### 3.5.2. Experimental Process

It is essential to use the experiment design program for settings prior to the experiment so that the visual stimuli can be represented on the VR equipment. We ran Experiment Center 4.0 to import the 30 preset visual stimuli, and inserted the description of the experiment, the black screen between stimuli, and the white screen for measuring the background EEG. We set the duration for exposing each content inserted, as well as the task to have each experimental group exposed. Following the completion of preparation, we performed a pilot experiment with 10 participants to check whether the participants would feel any discomfort with the simultaneous VR-EEG measurement, any incoming artifacts and noises were present, and data were seamlessly obtained. Moreover, we interviewed the participants to determine the appropriateness of the duration, for which the participants were exposed to the visual stimuli.

The experiment comprised three phases: (i) Pretest survey, (ii) EEG measurement, and (iii) posttest survey.

(i) Pretest survey: When participants arrived in the preparation room, we informed them of the purpose, process, and precautions of the experiment, as well as their right to drop out of the experiment. Furthermore, we double-checked if the participants had no problem with the EEG experiment using the questionnaire. Then, the participants were asked to fill in the questionnaire, which included the general information and the PWI-SF (PWI-Short Form).

(ii) EEG measurement: Upon the completion of the survey, the participants moved to the experiment room, sat on chairs comfortably, and took a break. After taking a sufficient break, they wiped their foreheads and earlobes before wearing the EEG equipment. Due to the COVID-19 pandemic, both researchers and participants were wearing face masks throughout the experiment. For the accuracy of measurements, we adjusted the headsets the participants were wearing and pressed each electrode to make sure it was attached to the scalp. When the impedance of all electrodes attached fell below 1, we checked if the participants felt any discomfort with the VR equipment they were wearing. Finally, we double-checked the impedance of electrodes, and examined the status of the EEG data recording from the participants wearing the equipment. We double-checked the presence of any incoming noises and the stability of EEG before starting the experiment.

As shown below, 19 channels of EEG electrodes were attached in accordance with the international 10/20(10–20%) system (Fp1, Fp2, Fz, F3, F4, F7, F8, Cz, C3, C4, T3, T4, T5, T6, P3, P4, Pz, O1, and O2). The reference electrode was attached to the back of the right earlobe, while the ground electrode was attached to the back of the left earlobe. The EEG was measured with the monopolar derivation.

Participants were asked to minimize their body movement and keep silent for the duration of the experiment. Still, they were allowed to turn their head to left and right naturally while experiencing the VR space. The experiment was designed to naturally flow following the description displayed in the VR. The overall experimental process is shown in Figure 5.

Firstly, while the participants were looking at the white screen visible in the VR in a comfortable state without any stimuli, the 1-min background EEG was measured. From the background EEG, the prestimulation RAB indicator values were extracted. The description of the overall experiment was presented before the participants experienced some similar visual stimuli to adapt to the VR space.

The 30 visual stimuli were grouped into six blocks, each of which included five stimuli combining the architectural elements, i.e., aspect ratios of space, ceiling heights, and window ratios (Figure 6). We set the duration of the exposure of the VR stimuli based on previous studies and the interview with the participants after the pilot experiment. The duration of the exposure of the EEG stimuli varies across authors. Yasuhisa et al. (2020) exposed each image stimulus for 6 seconds [67], whereas Wei-Yin et al. (2019) presented 60 photos at random for 20 seconds [68]. Agnieszka et al. (2018) presented 36 images at random for 10 seconds [41]. In this study, we interviewed the participants in the pilot experiment and found it took them 10 s to perceive a space without dozing off. Therefore, we chose to expose each stimulus for 10 s based on the interview results. Each block was run in the following order: The display of the introductory description—black screen (5 s)-stimulus 1 (10 s)-stimulus 2 (10 s)-stimulus 3 (10 s)-stimulus 4 (10 s)-stimulus 5 (10 s)-the display of the end-of-experiment statement—recess. Each block and the stimuli therein were arranged in random order. This process was repeated six times before the experiment ended. Each participant was exposed to the VR space for 9 min in total.

(iii) Posttest survey: Following the VR-EEG experiment, we surveyed the participants to compare the EEG response with the survey results about their consciousness. We asked the respondents to recall the space they had just experienced while filling in the questionnaire. The questionnaire comprised the items concerning the importance (1: not important at al~5: very important) of the architectural elements that influenced the relaxation-arousal responses following the VR experience and those concerning the consciousness of the relaxation and arousal in response to the images of the space they experienced (1: very arousing~5: very relaxing). Here, for the question items concerning the consciousness related to relaxation and arousal, we did not use the images of the entire experimental space. Instead, we used only 10 images of the environment whose ceiling height was 2.3 m and whose aspect and window ratios were varied, since the participants had difficulties in accurately perceiving and discriminating among the three different ceiling heights in the pilot experiment and interview. 

### 3.6. EEG Data Collection and Processing

Among the EEG data collected through the 19 channels, we analyzed the EEG data from 8 lobes, i.e., prefrontal (Fp1 and Fp2), frontal (F3 and F4), parietal (P3 and P4), and occipital lobes (O1 and O2) (Figure 7). The subjects’ EEG signals received from the 19 channels underwent the 16 bit AD (Analog-Digital) converter at 300 Hz sampling frequency, and was stored on the computer in the frequency passband of 0.003~150 Hz. We collected the measured EEG signals (raw data) using real-time data-collection SW DSI-streamer (ver 2.3, Wearable Sensing, City State Abbreviation, United States of America), converted the CSV file output using the time-series analysis program TeleScan (Ver 3.2, Laxtha, Daejeon, Korea), and analyzed the EEG data. We excluded 2 participants’ data containing lots of noises from the analysis, and used FFT (Fast Fourier Transformation) to convert the raw data and extract the RAB (Ratio Alpha/Beta). With 2 participants excluded, we statistically analyzed the EEG data of 31 out of 33 participants.

### 3.7. Statistical Analysis

EEG data do not directly lead to the overall pre- and poststimulation assessments. Moreover, it is not possible to directly read the emotional or relaxation-arousal states from EEG data [8]. Therefore, EEG data need to undergo processing and coding for statistical analysis and verification. To that end, we used SPSS 18.0 (IBM, Armonk, NY, USA) for coding the means of the participants’ pre- and poststimulus EEG data, and performed the Shapiro Wilk test for normality prior to the statistical analysis. The test results showed some EEG data had a *p*-value less than 0.05 and failed to secure the normality, which indicated a nonparametric statistical method should be used to analyze the EEG data collected in this study.

To test the null hypotheses set up earlier, we used the Wilcoxon signed-rank test, which is a nonparametric alternative to the paired sample t-test. The Wilcoxon signed-rank test is used to test if differences are present between *n* paired samples by ranking the observation scores of two samples. The equation used to yield the Wilcoxon test scores (W) is shown in Figure 8. Here, the significance level was *p* < 0.05.

## 4. Analysis Results

### 4.1. Questionnaire Survey Results

Vecchiato et al. [38], Banaei et al. [14], and Küller et al. [69] proved the correlation between the self-reported emotional response and the EEG response, and suggested the method of recognizing emotions based on vital sign measurements could be used as an objective tool for space ratings. This study also administered self-report questionnaire on the users’ awareness after experiencing VR room, with aiming to examine whether the results of the questionnaire survey and EEG are consistent with each other. The questionnaire consisted largely of questions about the importance of architectural elements applied to the virtual reality experience space, and the relaxation and arousal felt in the space with 2.3 m ceiling height applied.

The analysis of the postexperiment survey responses concerning the importance of the architectural elements (the aspect ratios of space, ceiling heights, and window ratios) in terms of inducing comfort highlighted the following. The aspect ratio of space and the window ratio averaged 4.06 and 4.77, respectively, in the analysis. The participants regarded those two ratios as important elements that could induce comfort. In contrast, the ceiling height averaged 3.77, whose importance was rated lower than the aspect and window ratios.

With regard to participants’ consciousness of relaxation and arousal in response to the varying aspect and window ratios in the environment where the ceiling height was 2.3 m, the survey results highlighted the following. The participants responded the window ratios of 60% and 80% in Type A and 60% and 40% in Type B were the elements that made them feel comfortable. In brief, the window ratio of 60% induced the most relaxation in both types, whereas the window ratio of 20% was found to induce the most arousal (Table 3, Figure 9).

### 4.2. Verifying the Differences in RAB Indicators in Response to Architectural Elements and Analyzing Activated Brain Lobes

#### 4.2.1. Aspect Ratio of Space—Type A

We used the Wilcox signed-rank test to compare the pre- and poststimulation RAB indicator values and to analyze the relaxation-arousal responses. In both Type A and Type B aspect ratios of space, statistically significant differences were found in response to the window ratios for each ceiling height (*p* < 0.05). That is, this result substantiates the effects of the changes to architectural elements on the EEG relaxation-arousal responses. First, the analysis of Type A found the following.

Significant differences in the pre- and poststimulation RAB indicator values were observed in Channels F3, F4, P3, and P4 for the 2.3 m ceiling height, in F4 and P3 for the 2.7 m ceiling height, and in Fp1, F3, F4, P3, and O2 for the 3.0 m ceiling height.

Above all, Channels F4 (right frontal lobe) and P3 (parietal lobe) showed significant differences in response to more window ratios than the other channels for all ceiling heights. All the channels that showed significant differences were characterized by negative (−) indicator values. This finding is attributable to the arousal response resulting from the decrease in the RAB after the participants experienced the stimuli in the VR space.

Meanwhile, some channels were observed to show characteristic patterns compared to others with no statistically significant differences. The prefrontal channels Fp1 and Fp2 showed the most negligible changes in the pre- and poststimulation RAB indicator values compared to the other channels. O1 was the only channel that showed positive (+) values of the pre- and poststimulation indicators in response to all window ratios. This finding is ascribable to the relaxation response resulting from the poststimulation increase in the RAB (Figure 10).

#### 4.2.2. Aspect Ratio of Space—Type B

Compared to Type A, the channels showing significant differences in the RAB indicator values were found more sporadic in Type B. Channels FP1, F3, F4, P3, P4, and O2 showed significant pre- and poststimulation differences for both 2.3 m and 2.7 m ceiling heights, whereas FP1, F3, F4, and P3 showed significant differences for the 3.0 m ceiling height. For all ceiling heights, the significant differences found in Channel P3 (left parietal lobe) were concentrated on the window ratios. A significant difference, or a great arousal response was observed in Channel O2 for the 2.3 m ceiling height.

As in Type A, all the indicator values of the channels showing the significant differences were negative (−) values, which confirmed the arousal response. Moreover, the relaxation response was observed in O1, which was the only channel that returned positive (+) RAB indicator values in response to all window ratios for the 2.3 m ceiling height.

The prefrontal lobes Fp1 and Fp2 showed significant differences in response to some window ratios and negligible changes in the pre and poststimulation RAB indicator values, compared to the other channels, as in Type A.

As a result, Channel P3 (left parietal lobe) showed statistically significant pre- and poststimulation differences in the RAB indicator values in both Types A and B (Figure 11).

### 4.3. Comparative Analysis of Arousal Levels Relative to RAB Indicator Values in P3

According to the analysis results discussed above, Channel P3 corresponding to the left parietal lobe overrode the other channels in terms of the significant differences in the pre- and poststimulation RAB indicator values. Furthermore, the channel was observed to have been most activated in response to the window ratios for each ceiling height in both Types A and B.

Based on the analysis of the differences in the pre- and poststimulation RAB indicator values in response to the changes to architectural elements, all the RAB indicator values that showed statistically significant differences were found to indicate the arousal response due to the decrease in the ratio of alpha to beta waves. However, as the observed arousal response varied across elements with different levels of arousal, the architectural elements showing the lowest arousal level in response to the VR space stimuli were derived by comparing the elements.

#### 4.3.1. Analysis of Arousal Levels Relative to Ceiling Heights—Aspect Ratios of Space Based on Window Ratios

First, as for the overall analysis results based on window ratios, when the window ratios were 20% and 40%, the changes in the RAB indicator values were not substantial across the aspect ratios of space and the ceiling heights. By contrast, when the window ratios were 60%, 80%, and 100%, the RAB indicator values substantially varied with the aspect ratios of space and the ceiling heights, leading to big differences in the arousal levels (Figure 12). This finding suggests that users’ arousal may substantially vary with the aspect ratios of space and the ceiling heights, should the window ratios of 60%, 80%, and 100% be applied to the architectural space design.

As for the window ratios relevant to the remarkable differences in arousal levels, in the virtual space with the window ratio fixed at 60%, the 2.3 m ceiling height was found to cause the least arousal response compared to the other ceiling heights in Type A. Conversely, this finding implies that the Type A. aspect ratio of space induced the least arousal response in the virtual space with the 2.3 m ceiling height. In Type B, the 3 m ceiling height was found to induce the least arousal response compared with the other ceiling heights. In the virtual space with the window ratio fixed at 80%, the 2.7 m and 3.0 m ceiling heights induced the least arousal response in Types A and B, respectively. In the virtual space with the window ratio fixed at 100%, the 2.7 m and 2.3 m ceiling heights induced the least arousal response in A and B, respectively (Table 4).

#### 4.3.2. Analysis of Arousal Levels Relative to Ceiling Heights-Window Ratios Based on Aspect Ratios of Space

The analysis results of the aspect ratios of space illuminated explicit differences 604 between Types A and B. First, the window ratio of 60% induced the least arousal response 605 when the ceiling height was 2.3 m in Type A. When the ceiling heights were 2.7 m and 3.0 m, the window ratio of 80% induced the least arousal response (Table 5, Figure 13).

In Type B, when the ceiling heights were 2.3 m and 2.7 m, the window ratios of 100% and 80% induced the least arousal response, respectively. The window ratios, induced the least arousal response when the ceiling heights, were 2.3 m and 2.7 m, whereas the window ratios of 40%, 60%, and 100% induced comparable arousal responses when the ceiling height was 3.0 m (Table 6, Figure 14).

### 4.4. Comparison of Survey Results and RAB Indicator Values

The survey results about the VR space, where the ceiling height was 2.3 m and the window ratios were varied, were compared with the EEG response.

All the significantly different RAB indicator values were relevant to the arousal response, and not completely consistent with the survey results that showed the relaxation response in some elements. Moreover, the survey results showed the comparable relaxation-arousal consciousness of the window ratios between Types A and B aspect ratios of space (see Figure 15), whereas the EEG results were different between the two aspect ratios of space.

Still, some similarities were also found between the survey results and the EEG response. Specifically, the survey respondents felt most comfortable with the window ratio of 60%, which indicated that the relaxation response dominated. Likewise, the analysis of the EEG response found the window ratio of 60% induced the least arousal response in Type A (2.3 m).

## 5. Discussion

The purpose of this study is to identify the effects of varied architectural elements on human brain response, and thereby explore if it is possible to determine the architectural elements that exert positive effects.

From the verification of the hypotheses set up in the introduction section, the key findings emerge as follows.

First, as for the statistical analysis of the hypothesis <HO 1>, the hypothesis was because statistically significant differences were observed in the pre- and poststimulation EEG-based RAB indicator values, while the participants were experiencing the VR space where architectural elements were varied. To begin with, we analyzed the differences in the pre- and poststimulation RAB indicator values in response to the VR space where the aspect ratio of space and the ceiling height were applied to the VR visual stimuli. As a result, significant differences were observed in both Types A (1:1.6) and B (1.6:1), while at the same time, the greater arousal response was observed in B than A. The channels that showed significant differences varied with the ceiling height elements. Still, the comparative analysis of the ceiling heights (2.3 m, 2.7 m, and 3.0 m) did not find any substantial changes in the indicator values. That is, any evident relevance of the RAB indicator values to the higher or lower ceiling heights was not found.

The findings in this study suggest the aspect ratios of space could influence the changes in the alpha and beta waves when the area of space is equal. That is, the aspect ratio of space may possibly exert a positive or negative influence on users’ physiological response. This finding should be noted in reference to Arthur E.S. [70,71] that used psychological assessments. Arthur E. S. conducted a study using a VR distance model and demonstrated that it is possible to increase the breadth of the awareness of the area of a space by modifying the shape of the space and its aspect ratio even when the actual area of the space is fixed. Moreover, a study on the interior space of guest rooms proved the effects of the aspect ratios (width-length 1:1, 1:2, 1:9) on users’ awareness of the area of a space exceeded the influence of the horizontal area and height elements [72]. Although the foregoing study results based on surveys and incomparable topics do not fully support the results of this study, they are in line with the present findings on the grounds that they verified the effects of the ratios of space on occupants’ cognitive and psychological response.

The analysis of the window ratio, which is the other architectural element applied to the experiment, found significant differences in the pre- and poststimulation RAB values. In particular, unlike the analysis results about the aspect ratios of space ceiling heights, the sizes of the changes in the indicator values were different from those in the channels where significant differences were observed across the ceiling heights. Ulrich, R. S. [60] scientifically proved the effects of the view seen from the windows of hospital rooms. Barbara M [61]. verified the shapes of windows influenced the perceived room sizes in a small area. Thus, the effects of windows on the human cognitive and physiological response have been well-documented. However, this study employed the VR and EEG methods and verified the effects of the window ratios on the EEG relaxation-arousal responses, adding to the existing body of knowledge.

Notably, the differences in the pre- and poststimulation RAB indicator values showed the arousal response in all channels except Fp1, Fp2, and O1. According to Maryam Banaei et al. [14], the designed architectural environment in virtual reality proved good enough to support the feelings of being in a real environment. Nevertheless, the rationale for the overall arousal response found in this study needs to be explicated. Heo et al. [73] reported that the equipment used in VR was characterized by the lenses that were close to the eyes and enlarged the positions and screens of displays, and induced the activation of the cognitive and functional neural network. They articulated a negative view that such characteristics masked the statistically significant differences in the results. Filip Škola et al. [9] measured the EEG response of users who were experiencing a 360° immersive VR application program, and reported that a high beta band substantially increased in the later phase of EEG measurement. They deduced the participants’ VR experience raised their cognitive processing. Additionally, wearing the EEG and VR equipment simultaneously may have added to the buildup of pressure. Thus, when it comes to research on users’ relaxation-arousal responses, VR technology needs to be verified by comparing the given results with those gained from the experiments in real settings, despite the fact that VR constitutes efficient tools for representing the architectural space. Nonetheless, we obtained the data applicable to the design in practice based on the analysis findings on account of the distinctive arousal levels. In the same vein, identifying the architectural elements causing the least arousal response in a setting can provide important clues for architectural design. When using VR as a tool for representing the architectural space, it would be necessary to minimize the physical discomfort of participants by controlling the duration of VR stimuli and the recess. Moreover, as Vecchiato et al. [38] pointed out, 3D stimulus materials which allow users to turn their head and torso are a remarkable advancement towards more substantive measurements, but still, they lack in a natural movement by way of the implemented environment. Therefore, a method of combining the MoBI (Mobile Brain/Body Imaging) proposed by Maryam Banaei et al. [14] with VR is worth considering.

Second, as for the statistical analysis of the hypothesis <HO 2>, the hypothesis was rejected since certain brain lobes showed statistically significant differences in the relaxation-arousal responses following the experience in the VR space.

The brain lobes that showed significantly different pre- and poststimulation RAB indicator values varied across the types of the aspect ratios of space and the ceiling heights. Multiple significant differences were found in Channels F4 (right frontal lobe) and P3 (left parietal lobe) in Type A, and P3 (left parietal lobe) in Type B. The present findings are supported by the literature described below.

Vartanian et al. [58] recorded the brain wave signals of 12 participants, who were recognizing three immersive VR settings. They proved the alpha band was activated in the left-center parietal lobe and frontal lobes devoted to the visual spatial exploration and the categorial spatial relation processing in the VR environment, which was reported by the participants as an exciting space. The relevant structures of the two areas play important roles in visual spatial processing. The categorical spatial relations involve the tasks in an environment where users explore a limited VR space, as they did in the present study, that does not require accurate positions [74]. As the visual exploration in VR spaces does not require the processing of certain distances between objects, it is reasonable to assume that the subjects’ categorical spatial processing was activated while they were recognized the agreeable environment [58] From a different viewpoint of the parietal lobe, Monaco et al. [75] discussed that the parietal lobe was involved in the integration of information about 3D objects in relation to identifying the objects and their sizes. Juha et al. [76] demonstrated that the activities of the parietal lobe relevant to goal-oriented actions varied with the priorities of actions that explained the perception, thoughts, and emotions, while the subjects were observing natural scenery. Finally, Baciu et al. [77] and Suegami et al. [78] reported the functions of the left parietal lobe largely involve the processing of spatial relations.

As a result, the present findings on the significant response of the right frontal lobe and the left parietal lobe to EEG in the VR environment, where the architectural elements were applied, support the literature. Our findings also support the need to carefully observe the activation of these two regions while measuring the brain response to space. Moreover, as the only channel that showed the relaxation response, which was not statistically significant, Channel O1 (left occipital lobe) must be discussed in further studies.

Third, to analyze the EEG response and the self-report survey results, Vecchiato et al. [38], Banaei et al. [14], and Küller et al. [69] proved the correlation between the self-reported emotional response and the EEG response, and suggested the method of recognizing emotions based on vital sign measurements could be used as an objective tool for space ratings. Ryu analyzed the correlation between the EEG indicators and the survey results about the vocabulary used to rate residential spaces in line with the changes of colors, and reported that the emotional response to the changes of colors in each residential space was largely comparable to the EEG response and that the two responses were less correlated in 17 subjects, which the author considered attributable to personal experience and disposition. In this study, we conducted the self-report survey about the relaxation-arousal responses to the 2.3 m-ceiling space following the VR experience. The comparative analysis of the survey results and the EEG response found the self-reports and the relaxation-arousal responses did not completely match.

This finding warrants the need for follow-up research on the themes of this study on the following two grounds.

First, drawing solely on the self-report psychological assessment to rate spaces is likely to cause error-prone results. Therefore, the objectivity of results should be secured by measuring the physiological response in addition to the EEG response and the psychological assessment with surveys or interviews and by comparatively analyzing those results. In this study, as the participants filled in the questionnaire upon the completion of the EEG experiment, the interval between the two points of time may have hindered their consciousness of the VR experience from being reflected as is, and the likelihood that the time lapse led to the differences between the survey results and the EEG response cannot be ruled out. Hence, follow-up studies should design the experiment so that each experimental block of the architectural elements is immediately followed by a survey. Furthermore, this experiment analyzed changes in brainwave responses before and after stimulation for the same group with similar stress levels, but it is necessary to scrutinize the socio-demographic characteristics of future participants and analyze them for differences in brainwave responses between groups.

Lastly, the implications for practitioners when they design private rooms in postpartum care centers are presented based on the analysis result of arousal level measured using RAB indicators.

The EGG analysis showed that when the window ratios are 20 and 40%, the aspect ratios and ceiling height had no strong effect on users’ arousal level, while, when the ratios are 60, 80, and 100%, both variables had a strong effect on users’ arousal level, and that there were significant differences in the arousal level induced by Types A and B. These results may provide a cue for the practitioners to design more comfortable space based on the users’ physiologic reaction. The design standards recommended for practitioners when they design private rooms in postpartum care center are as follows:First, the recommended ceiling heights for Types A and B by window ratios are 2.3 and 3.0 m for the ratio of 60%, 2.7 and 3.0 m for the ratio of 80%, 2.7 and 2.3 m for the ratio of 100%.Second, the combinations of ceiling height and window ratio that induced the lowest level of arousal reaction were 2.3 m and 60% for Type A (1:1.6) and 2.3 m and 100% for Type B (1.6:1), and accordingly, we recommend these combinations.

## 6. Conclusions

This study was intended to establish the effects of varied architectural elements on users’ relaxation-arousal responses by measuring the EEG response to the cognitive stimuli received from the space represented in immersive VR.

As a result, statistically significant differences were observed in the changes of the pre- and poststimulation EEG RAB indicator values, while the participants were experiencing the VR space where the architectural elements were varied. That is, we statistically verified the effects of the changes to architectural elements on users’ relaxation-arousal responses. Noteworthily, in all the RAB indicator values that showed significant differences, the poststimulation ratio of the alpha to beta waves decreased compared to the prestimulation ratio, which was indicative of the arousal response. However, the arousal levels varied across the architectural elements, which derived the elements that induced the less arousal response using this method.

Moreover, certain brain lobes showed statistically significant differences in the relaxation-arousal responses following the VR experience. Channels F4 (right frontal lobe) and P3 (left parietal lobe) showed multiple significant differences in the RAB indicators. In that, ns rath both regions of the brain are important to the visual spatial processing and known to be involved in the tasks of processing the categorical space, or the environment where subjects explore VR spaces, the findings of this study support the literature.

The EEG response and the survey results did not perfectly match, but some similarities were derived. This finding implies that any attempt to rate spaces should not entirely rely on self-reported psychological assessments and warrants the need for follow-up studies to establish the relationship between psychological and physiological assessments. This study performed EEG experiments with the participants restricted to females who had the experience of using private rooms in postpartum care centers. It is considered that the generalization of the result requires further studies on other healing spaces with participants, including general populations, rather than restricting research to the female population.

To make sure the RAB indicators and certain brain wave channels (F4 and P3) are used as the biomarkers conducive to analyze users’ relaxation-arousal responses in VR spaces, further studies should verify their applicability using larger samples.

## Figures and Tables

**Figure 1 ijerph-18-04305-f001:**
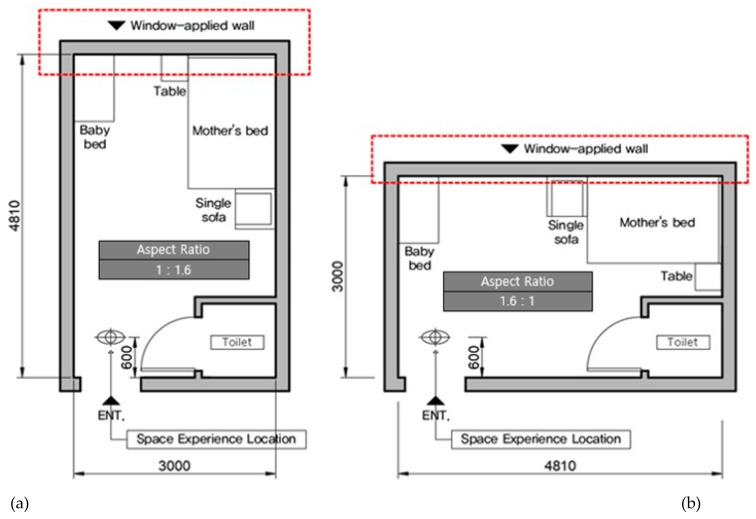
Floor Plan of space reproduced in virtual reality (VR). (**a**) Aspect ratio A type (1:1.6). (**b**) Aspect ratio B type (1:1.6). A change in the window area ratio was applied to the wall inside the red square box. The scene seen during the VR experience is the front when you enter the door. When measuring brain waves at this point, the head can be moved slowly from side to side to observe the space.

**Figure 2 ijerph-18-04305-f002:**
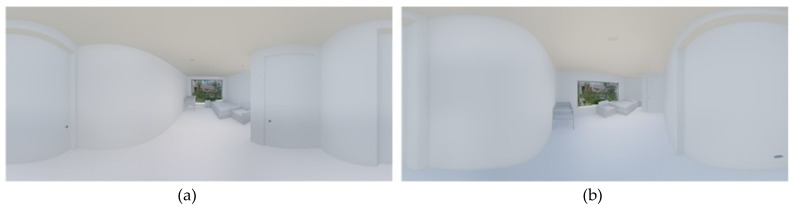
VR space 360° 3D image example. (**a**) It is an image with 2.3 m of ceiling height and 80% of window area ratio applied to the aspect ratio A type of space. (**b**) It is an image with 2.3 m of ceiling height and 60% of window area ratio applied to the aspect ratio of the space B type.

**Figure 3 ijerph-18-04305-f003:**
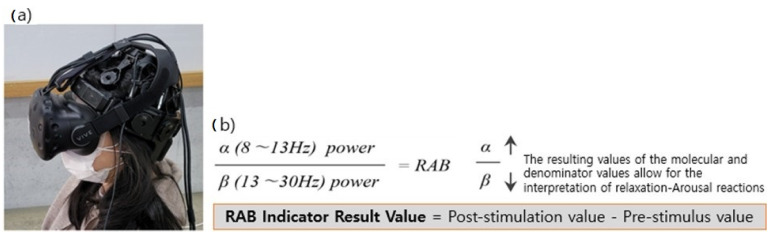
(**a**) Wearing experimental equipment. EEG equipment was first worn, and VR equipment was worn on top of it. (**b**) The ratio of alpha to beta waves (RAB) indicators and formula for deriving results. If the RAB indicator results presented in this paper are positive (+) above zero, it can be interpreted that the proportion of alpha waves to beta waves increased after stimulation than before and after stimulation, resulting in relaxation. A negative (−) value can be interpreted as a reduced proportion of alpha waves to beta waves, resulting in an arousal response.

**Figure 4 ijerph-18-04305-f004:**
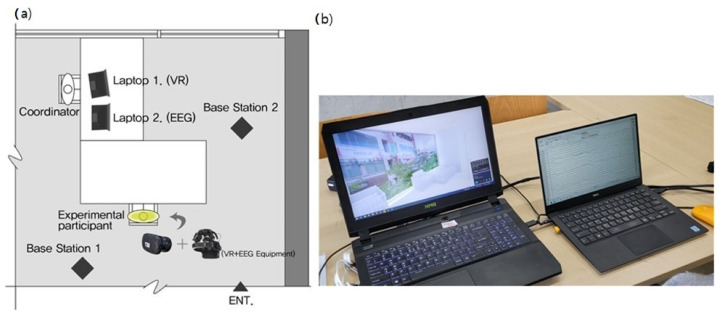
(**a**) Laboratory schematic diagram. The controller operated two laptops in front of the two pieces of equipment. The base station required for VR to operate is installed so that it faces diagonally around the experimental participants and synchronizes it. (**b**) EEG-VR adjustment laptops. The laptop on the left is for VR adjustment, and researchers can see where the experimental participants are looking in real-time by moving their eyes in the VR space. The laptop on the right sets the brain wave measurement time and data storage status for EEG adjustment, and can observe changes in brain waves in real-time.

**Figure 5 ijerph-18-04305-f005:**
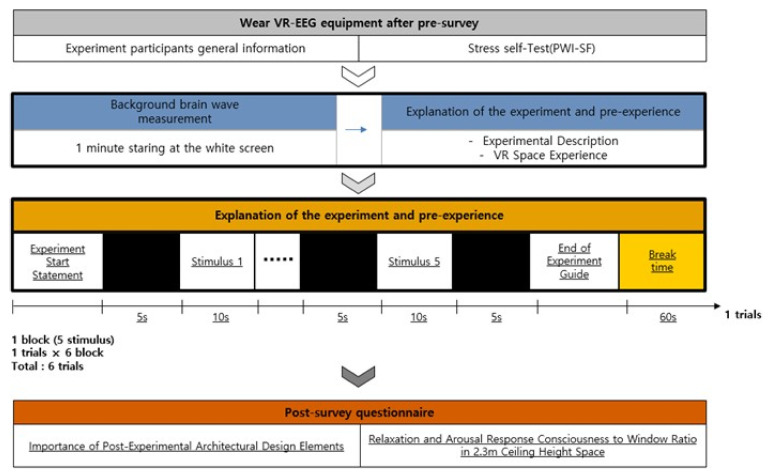
The procedure of an experiment.

**Figure 6 ijerph-18-04305-f006:**
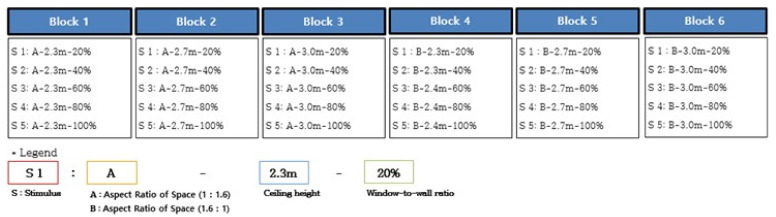
VR visual stimulation.

**Figure 7 ijerph-18-04305-f007:**
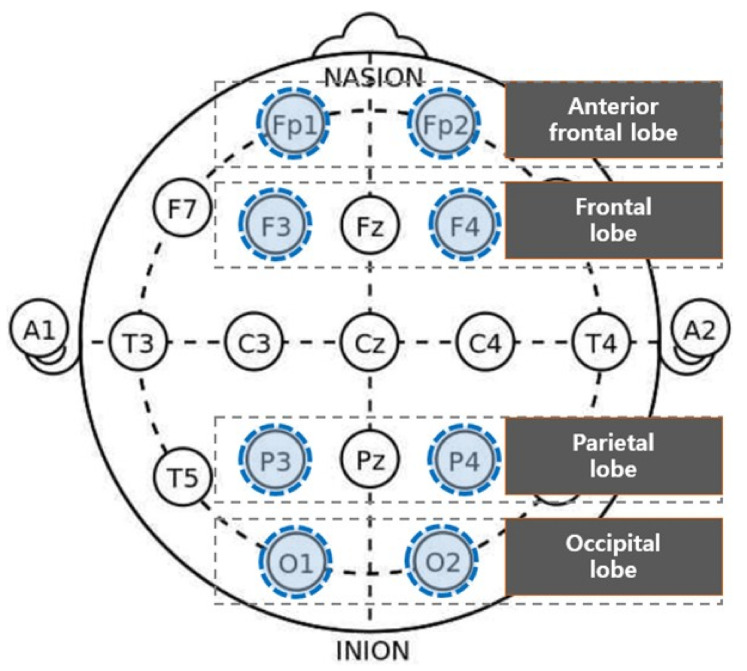
Brain wave measuring area.

**Figure 8 ijerph-18-04305-f008:**
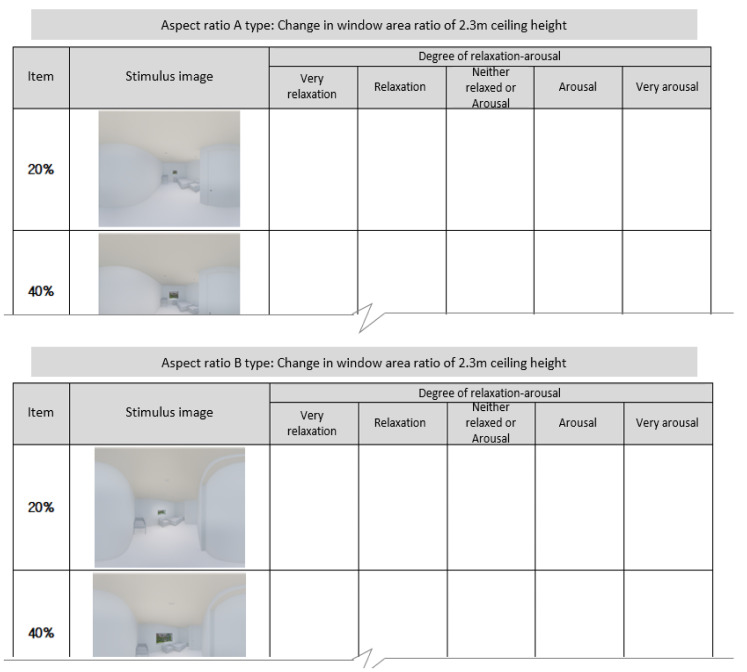
Relaxation-arousal questionnaire.

**Figure 9 ijerph-18-04305-f009:**
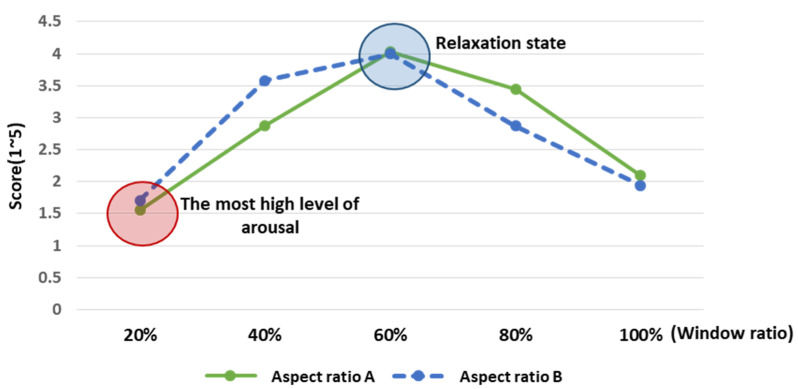
Analysis of relaxation-arousal levels for window ratio by aspect ratio. An average of four points or more were felt relaxed in the corresponding element. The window ratio, shown in the relaxation state, was marked with a blue circle, while the window ratio, shown in the highest arousal state, was marked with a red circle.

**Figure 10 ijerph-18-04305-f010:**
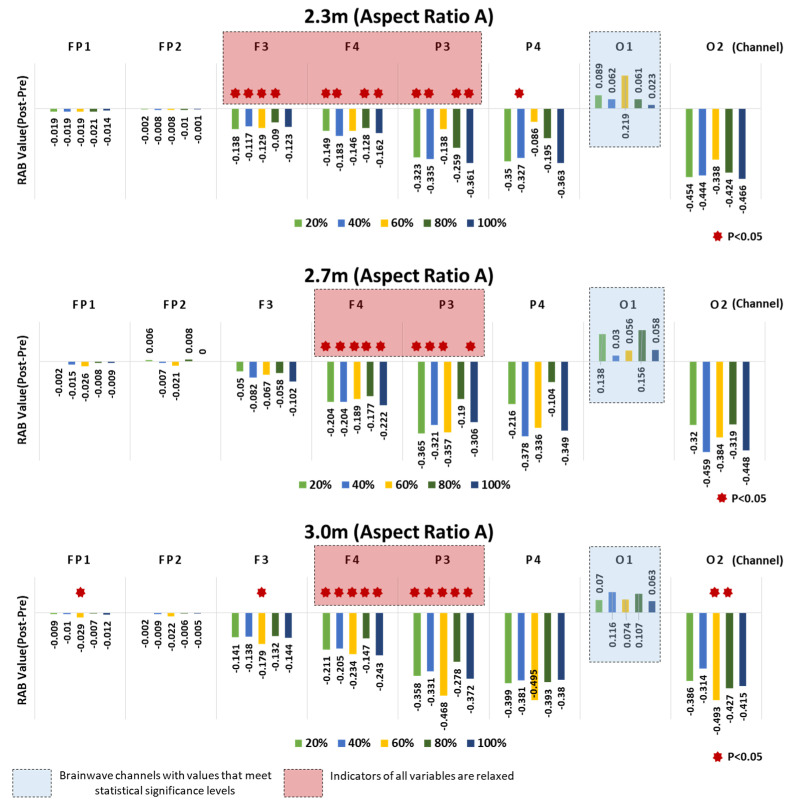
Aspect Ratio A—results of RAB indicator analysis for window ratio by ceiling height type.

**Figure 11 ijerph-18-04305-f011:**
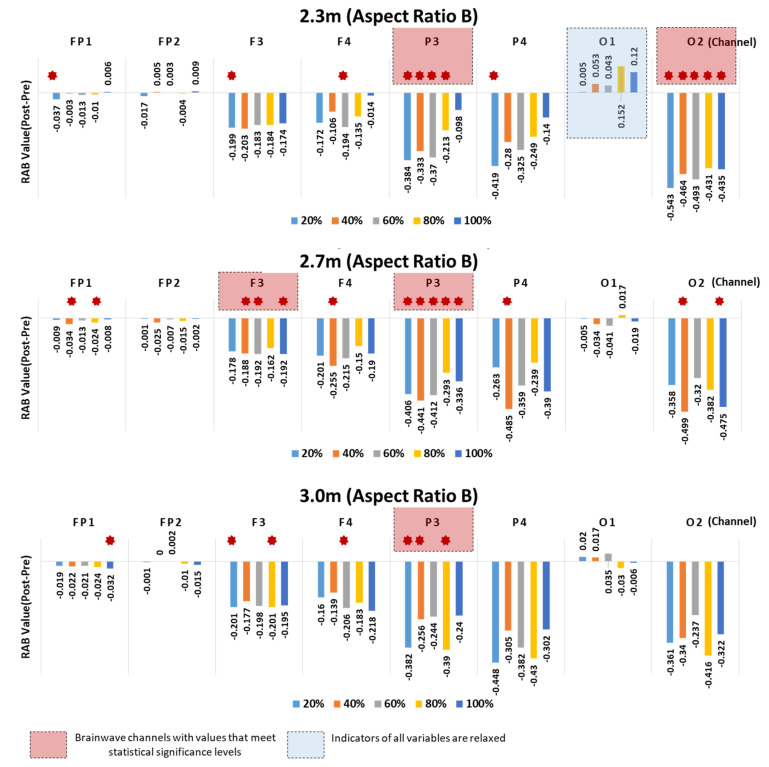
Aspect Ratio B—results of RAB indicator analysis for window ratio by ceiling height type.

**Figure 12 ijerph-18-04305-f012:**
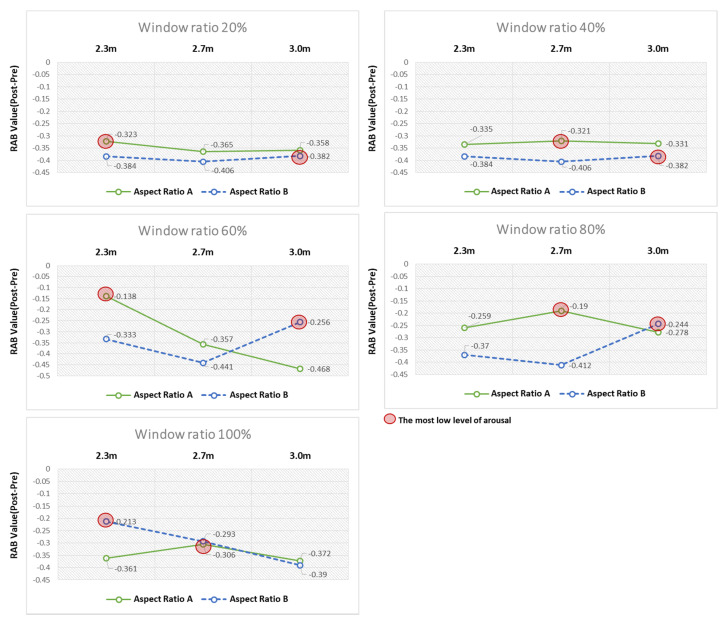
Analysis of arousal levels based on window ratio. As a result of the comparison between elements of ceiling height, the element with the smallest level of awakening was marked with a red circle.

**Figure 13 ijerph-18-04305-f013:**
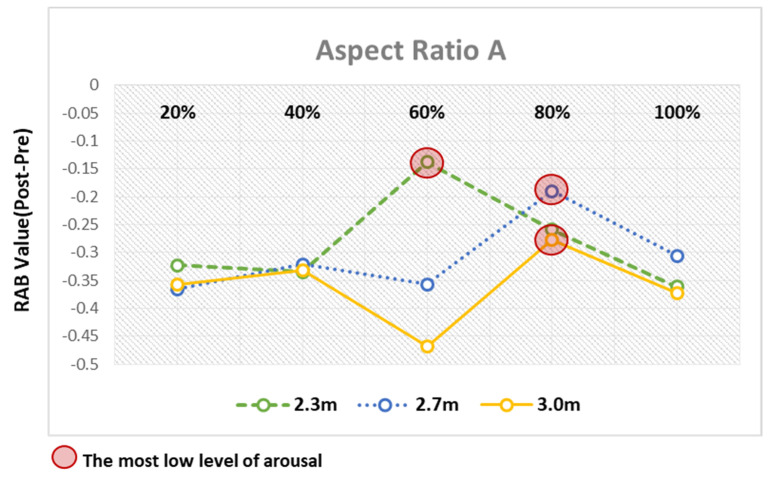
Analysis of arousal levels based on aspect ratio A.

**Figure 14 ijerph-18-04305-f014:**
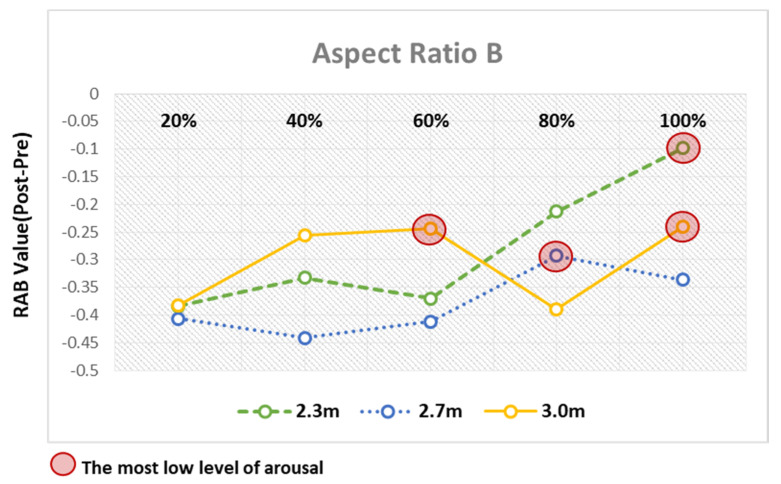
Analysis of arousal levels based on aspect ratio B.

**Figure 15 ijerph-18-04305-f015:**
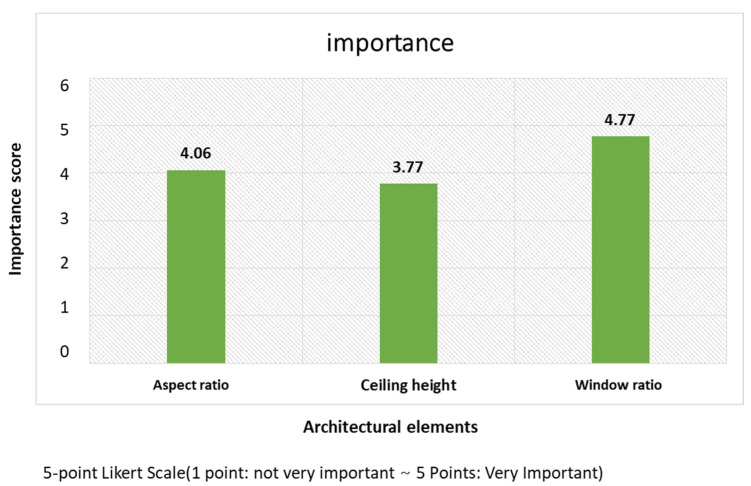
Importance analysis of architectural elements. This is the result of a survey on the importance of architectural elements affecting relaxation and arousal.

**Table 1 ijerph-18-04305-t001:** Intergroup homogeneity test for background EEGs.

Item	Mean		SD		*t*-Value	*p*-Value
	Health Group (*n* = 6)	Potential Stress Group (*n* = 25)	Health Group (*n* = 6)	Potential Stress Group (*n* = 25)		
Fp1	0.109	0.115	0.069	0.087	−0.143	0.888
Fp2	0.112	0.099	0.081	0.076	0.384	0.704
F3	0.740	0.261	1.180	0.307	0.985	0.368
F4	0.656	0.385	1.363	0.511	0.479	0.651
P3	1.018	0.976	1.228	1.002	0.088	0.930
P4	1.041	1.039	1.525	1.527	0.003	0.998
O1	0.367	0.358	0.352	0.265	0.075	0.941
O2	0.318	1.043	0.419	2.266	−0.771	0.447

Health groups—8 points or less, Potential stress groups—9 to 26 points.

**Table 2 ijerph-18-04305-t002:** Participants’ general characteristics and stress scores (*n* = 33).

Item	Details	Frequency	%
Age	30 s	26	78.79
	40 s	7	21.21
Education	High school graduate	1	3.03
	Bachelor	27	81.82
	Master/PhD	5	15.15
Occupation	Student	1	3.03
	Housewife	23	69.70
	Employee	8	24.24
	Self-employed	1	3.03
Used postpartum care centers	<1 year ago	5	15.15
	1~2 years ago	2	6.06
	2~3 years ago	11	33.33
	3~4 years ago	5	15.15
	4~5 years ago	4	12.12
	5~10 years ago	6	18.18
PWI-SF stress scores	≤8 (healthy group)	6	18.18
	9~26 (potential stress group)	27	81.82

**Table 3 ijerph-18-04305-t003:** Survey on relaxation-arousal consciousness in response to window ratios per aspect ratio of space (*n* = 33).

Item		Mean (M)	SD	Ranking
Aspect Ratio	Window Ratio			
Type A	20%	1.55	0.675	5
	40%	2.87	1.118	3
	60%	4.03	0.836	1
	80%	3.45	1.091	2
	100%	2.10	0.978	4
Type B	20%	1.71	0.938	5
	40%	3.58	1.025	2
	60%	4.0	0.894	1
	80%	2.87	1.204	3
	100%	1.94	1.124	4

Legend. 5-point Likert scale (1: very arousing ∼5: very relaxing).

**Table 4 ijerph-18-04305-t004:** Analysis of the difference in RAB indicator values between ceiling height and aspect ratio (A/B).

Channel	2.3 m				2.7 m				3.0 m			
	A Type		B Type		A Type		B Type		A Type		B Type	
	Difference	*p*- Value	Difference	*p*- Value	Difference	*p*- Value	Difference	*p*- Value	Difference	*p*- Value	Difference	*p*- Value
20%	−0.323	0.009 *	−0.384	0.001 *	−0.365	0.000 *	−0.406	0.001 *	−0.358	0.002 *	−0.382	0.000 *
40%	−0.335	0.007 *	−0.384	0.001 *	−0.321	0.025 *	−0.406	0.001 *	−0.331	0.002 *	−0.382	0.000 *
60%	−0.138	0.126	−0.333	0.004 *	−0.357	0.004 *	−0.441	0.003 *	−0.468	0.002 *	−0.256	0.036 *
80%	−0.259	0.020 *	−0.37	0.034 *	−0.19	0.117	−0.412	0.002 *	−0.278	0.001 *	−0.244	0.170
100%	−0.361	0.028 *	−0.213	0.038 *	−0.306	0.044 *	−0.293	0.028 *	−0.372	0.014 *	−0.39	0.014 *

* *p* < 0.05.

**Table 5 ijerph-18-04305-t005:** Analysis of difference between ceiling height and window ratio RAB indicators of aspect ratio A.

Index	20%		40%		60%		80%		100%	
	Difference	*p*-Value	Difference	*p*-Value	Difference	*p*-Value	Difference	*p*-Value	Difference	*p*-Value
2.3 m	−0.323	0.009 *	−0.335	0.007 *	−0.138	0.126	−0.259	0.020 *	−0.361	0.028 *
2.7 m	−0.365	0.000 *	−0.321	0.025 *	−0.357	0.004 *	−0.19	0.117	−0.306	0.044 *
3.0 m	−0.358	0.002 *	−0.331	0.002 *	−0.468	0.002 *	−0.278	0.001 *	−0.372	0.014 *

* *p* < 0.05.

**Table 6 ijerph-18-04305-t006:** Analysis of difference between ceiling height and window ratio RAB indicators of aspect ratio B.

Index	20%		40%		60%		80%		100%	
	Difference	*p*-Value	Difference	*p*-Value	Difference	*p*-Value	Difference	*p*-Value	Difference	*p*-Value
2.3 m	−0.384	0.001 *	−0.333	0.001 *	−0.37	0.034 *	−0.213	0.038 *	−0.098	0.092
2.7 m	−0.406	0.001 *	−0.441	0.001 *	−0.412	0.002 *	−0.293	0.028 *	−0.336	0.034 *
3.0 m	−0.382	0.000 *	−0.256	0.000 *	−0.244	0.17	−0.39	0.014 *	−0.24	0.272

* *p* < 0.05.

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
