# Peer review of "Effects of Changes to Architectural Elements on Human Relaxation-Arousal Responses: Based on VR and EEG"

_ijerph, 2021, doi:10.3390/ijerph18084305_

Round 1
Reviewer 1 Report
This is a very interesting study on the influence of architectural elements on the occupants’ stress-related physiological changes. The study employs both experiments as well as self-reported survey data. The experiment design and data analysis seem solid and the findings are interesting as well. It is believed that the manuscript is highly potential to contribute to the research community and industry. Especially, the findings can contribute to the built environment design process. Here are some comments:
- The abstract is very informative. It presents the methodology, experiment design as well as results in detail. However, I believe the abstract might be too long for readers: they expect to find the main points of the article within a few minutes and have a general understanding of the research method and results. In view of the open-access of IJERPH, authors do not need to worry about whether readers have the opportunity to read the details described in the main text. Therefore, I suggest that the author provide a more condensed summary to increase readability. Some details in the abstract can be placed in the introduction and summary.
- The authors are highly suggested to provide more information on the hypotheses in section 1.2. Please explain “Why them”? Maybe a short literature review can be considered.
- Please explain the reason why only women participate in the experiment and if there is a risk of result bias. If so, the reviewer would like to have a limitation section before the conclusion and discuss this clearly.
- Section 4.1 might require more explanation. Figure 9 seems to work for Section 4.1. If so, please mark it there.
In addition to the above points, here are some minor points:
- Please do not use abbreviations (such as “EEG”) in the abstract.
- The authors are recommended to pay more attention to the uniformity of the citation format. For example, the authors widely employ the MDPI reference style while employ the APA style between line 58 to line 60.
- The authors are suggested to explain the meaning of “º” and “©” in line 126 or to delete them if not necessary.
- Please give the full name of “IRB” in the main body.
- The authors should not use a figure to present the formula (Figure 8).
- A higher-quality version of Figure 11-14 is necessary so that the readers can better understand the points.
- Line 608 mentioned that “Please see Figure 14 and 15” but only Figure 14 was found. The reviewer guesses that the figure on the right-hand side in Figure 15. Please mark it if so or provide Figure 15 if not.
Author Response
1. (Line_1-24)
A concise summary of the content that includes the purpose, methodology, results, and originality of the study.
2. (Line_240-262, 268-270)
(1.2 Scope of Study and Hypothesis) was moved to (3:3.1).
It was described in the text that a prior study on cognitive and physiological responses for architectural changes was added to the contents of Chapter 2, and hypotheses were established based on the results of prior research.
3. (Line_5, 107-112, 793-796)
The study space is the mother's room in the postpartum care center. We conducted an experiment on women who have experienced using this space, not ordinary people or students. Since the mother needs physical and psychological stability after giving birth, it is necessary to measure the brain wave response of the actual user and reflect it in the plan of this space. However, it is believed that subsequent studies of the public will be necessary to generalize the findings. This was described in (1.2 Scope of Study, 6. Conclusion).
4. (Line_501-510)
Figure 9 was positioned incorrectly during the editing process of editing. I modified this part.
We also supplemented the additional explanation for Section 4.1.
5. (Line_1,3)
The abbreviations used in abstract were also modified to describe the full name together.
6. (Line_36-40)
The citation format has been modified to fit the MDPI baseline style.
7. The authors are suggested to explain the meaning of “º” and “©” in line 126 or to delete them if not necessary.
-> We deleted it.
8. (Line_280)
The modification has been completed.
9. The authors should not use a figure to present the formula (Figure 8).
-> We deleted it.
10. (Line_10-13)
The high-quality version of Figure has been replaced
11. (Line_13-14)
The modification has been completed.
We revised the paper according to the review results. However, the line number does not go straight, so I will discuss this with the mdpi editor and correct it. Please review content-oriented. Thank you.

Reviewer 2 Report
The paper is very difficult to read as it is poorly structured and with unclear ideas. I mean, confusing The abstract is too long, unstructured and the objectives are not clearly specified. In the title and introduction they use abbreviations and initials without explaining the full name of the term used. It does not separate discussion from results. Anyway, they write it again from start to finishAuthor Response
1. (Line_1-24)
A concise summary of the content that includes the purpose, methodology, results, and originality of the study.
2. (Line_1,3)
The first abbreviation has been modified to describe the full name together.
3. (Line_626,770)
I revised it by dividing it into the conclusion of Chapter 6.
We revised the paper according to the review results. However, the line number does not go straight, so I will discuss this with the mdpi editor and correct it. Please review content-oriented. Thank you.

Reviewer 3 Report
Dear authors, I found the paper very interesting; nonetheless, there are some points you need to address:
a. the abstract is too long; it must be a concise summary including the aims of the study, the methodology, the results and originality of the study. Please try to avoid to provide all the results of the study, but just focus on the few of them, the more relevant for the journal's audience. Again, the part related to the methodology should be concise. In addition, you refer to a questionnaire, without explaining the reasons why you used it. EEG, RAB indicator and VR should be explained, and only after, followed by the acronym.
b. the section 1.2 Scope and Hypothesis should be moved after section 2.1 as hypotheses should be derived from literature, and should follow the indication from literature. So that, if your literature review reveals differences in EEG based relaxation-arousal responses when varying the architectural elements (using VR experience), you should expect a difference of perceptions in your study. Otherwise, you shoul state that you don't expect meaningful differences based on the previously conducted literature review.
You should do the same for hypothesis 2.
If studies emerging from previous literature are not sufficient to identify the directions of the hypotheses you should give this information, when introducing the 2 hypotheses.
c. the implications deriving from the study and related to the design of architectural spaces in healthcare should be clearly stated in the discussions section. How do practitioners can use your results for managerial purposes?
d. a professional proof reading should be provided. Some errors have been detected: for instance. line 251, 407,413 you should not capitalized after the colon.
Author Response
1. A concise summary of the content that includes the purpose, methodology, results, and originality of the study.
Also, the abbreviations used in abstract were also modified to describe the full name together.
We specified the reason why we used the questionnaire.
2. (1.2 Scope of Study and Hypothesis) was moved to (3:3.1).
It was described in the text that a prior study on cognitive and physiological responses for architectural changes was added to the contents of Chapter 2, and hypotheses were established based on the results of prior research.
3. The results of the study clearly stated the meaning related to architectural space design. It also described how working-level officials can utilize the research results in the future.
4. we corrected the part where the error was confirmed after calibration.
We revised the paper according to the review results. However, the line number does not go straight, so I will discuss this with the mdpi editor and correct it. Please review content-oriented. Thank you.

Round 2
Reviewer 1 Report
This is an interesting and highly promising study and may advance knowledge in architecture and building sciences. The revised version of this manuscript is significantly improved. It is also believed that the results of this study are clear and useful. Here are some comments as follows:
- The authors are recommended to compare the sociodemographic characteristics to the statistical data on the Daegu and also national scale in section 3.2. In addition, the manuscript only considers three sociodemographic factors (i.e., age, education, occupation). If it possible, they are highly suggested to add (family) income, personality characteristics and their partners’ characteristics and discuss them in section 3.2. If not, please do consider adding a limitation paragraph in the discussion and explain them in detail.
Here are some resources that might be useful:
National Symbols of the Republic of Korea, here is the link (for your information): http://kostat.go.kr/portal/eng/index.action
Here are also some highly-related and most updated papers for your reference and information:
Socio-demographic Description:
Wang, Q. C., Chang, R., Xu, Q., Liu, X., Jian, I. Y., Ma, Y. T., & Wang, Y. X. (2021). The impact of personality traits on household energy conservation behavioral intentions–An empirical study based on theory of planned behavior in Xi'an. Sustainable Energy Technologies and Assessments, 43, 100949.
Process Description:
Amihai, I., & Kozhevnikov, M. (2014). Arousal vs. relaxation: a comparison of the neurophysiological and cognitive correlates of Vajrayana and Theravada meditative practices. PloS one, 9(7), e102990.
- In addition, it would great if they can (1) adding one sentence between line 307 to 310 to explain how they classify the participants into two groups (although illustrated in Table 1), (2) compare the sociodemographic characteristics of two groups in Table 1 as well.
- The authors are highly recommended to introduce comprehensively and present the questionnaire they employed in the study (may be listed as an appendix).
- Follow the last version comments point 7. Please find that in the updated version, you did not delete the “º” and “©” in line 113 to 114.
Besides, as mentioned by other reviewers, the academic writing of this manuscript needs further improvement. The authors may consider extensive editing of English writing.
Good luck.
Author Response
- The authors are recommended to compare the sociodemographic characteristics to the statistical data on the Daegu and also national scale in section 3.2. In addition, the manuscript only considers three sociodemographic factors (i.e., age, education, occupation). If it possible, they are highly suggested to add (family) income, personality characteristics and their partners’ characteristics and discuss them in section 3.2. If not, please do consider adding a limitation paragraph in the discussion and explain them in detail.
-> (Line_310, 754)
- We know that a close review of participants' socio-demographic characteristics has a significant impact on the interpretation of the results of the brainwave response analysis.
- However, the study does not compare social demographic characteristics with gender-specific brainwave responses.
- Therefore, the participants' psychological and physical stress conditions were considered important at the time of the experiment, and the stress self-test was conducted.
- Although all the points you pointed out were not reflected, the homogeneity test of the participant groups separated by the results of the stress self-test confirmed that there were no differences between groups, and the results were supplemented by adding them to Section 3.2.
- And this is described in 5. Discussion section.
-
Add) I searched the national statistics portal site, but there was no statistical data on education and occupation in Daegu.
2. In addition, it would great if they can (1) adding one sentence between line 307 to 310 to explain how they classify the participants into two groups (although illustrated in Table 1), (2) compare the sociodemographic characteristics of two groups in Table 1 as well.
-> (Line_310)
Since this study is not a comparison of brainwave data between groups, it is considered to be misleading if the contents of Table 1 are divided into groups.
I'm sorry that I couldn't fully reflect your intention. However, as described above, we will present the results of the intergroup homogeneity test as a table.
3. The authors are highly recommended to introduce comprehensively and present the questionnaire they employed in the study (may be listed as an appendix).
-> (Figure 8)
The Relaxation-Awakening Consciousness Survey questionnaire is shown in Figure 8.
4. Follow the last version comments point 7. Please find that in the updated version, you did not delete the “º” and “©” in line 113 to 114.
-> (Line_113) There seems to have been an error in the system. I deleted it, but if this symbol appears again in the final paper, I will contact the society.
Thanks for the review. Thesis editing will be resolved by discussing with the mdpi editor. Thank you.

Reviewer 2 Report
I think the paper has improved in its current format and can be accepted
Author Response
Thanks for the review.
Thesis editing will be resolved by discussing with the mdpi editor.
Thank you.